# Effects of ACTH-Induced Long-Term Hypercortisolism on the Transcriptome of Canine Visceral Adipose Tissue

**DOI:** 10.3390/vetsci9060250

**Published:** 2022-05-25

**Authors:** Miguel Tavares Pereira, Isabelle Martin, Hubert Rehrauer, Mariusz P. Kowalewski, Felicitas S. Boretti, Nadja S. Sieber-Ruckstuhl

**Affiliations:** 1Institute of Veterinary Anatomy, Vetsuisse Faculty, University of Zurich, 8057 Zurich, Switzerland; miguel.tavarespereira@uzh.ch (M.T.P.); kowalewski@vetanat.uzh.ch (M.P.K.); 2Clinic for Small Animal Internal Medicine, Vetsuisse Faculty, University of Zurich, 8057 Zurich, Switzerland; imartin@vetclinics.uzh.ch; 3Functional Genomics Center Zurich, ETH Zurich, University of Zurich, 8057 Zurich, Switzerland; hubert.rehrauer@fgcz.ethz.ch

**Keywords:** dog (*Canis lupus familiaris*), Cushing’s syndrome, atherosclerosis, animal model

## Abstract

Cushing’s syndrome, or hypercortisolism (HC), a common endocrinopathy in adult dogs, is caused by chronic hypercortisolemia. Among different metabolic disorders, this syndrome is associated with enhanced subcutaneous lipolysis and visceral adiposity. However, effects of HC in adipose tissue, especially regarding visceral adipose tissue (VAT), are still poorly understood. Herein, the transcriptomic effects of chronic HC on VAT of dogs were evaluated. For this, subcutaneously implanted ACTH-releasing pumps were used, followed by deep RNA sequencing of the canine VAT. Prolonged HC seems to affect a plethora of regulatory mechanisms in VAT of treated dogs, with 1190 differentially expressed genes (DEGs, *p* and FDR < 0.01) being found. The 691 downregulated DEGs were mostly associated with functional terms like cell adhesion and migration, intracellular signaling, immune response, extracellular matrix and angiogenesis. Treatment also appeared to modulate local glucocorticoid and insulin signaling and hormonal sensitivity, and several factors, e.g., TIMP4, FGF1, CCR2, CXCR4 and HSD11B1/2, were identified as possible important players in the glucocorticoid-related expansion of VAT. Modulation of their function during chronic HC might present interesting targets for further clinical studies. Similarities in the effects of chronic HC on VAT of dogs and humans are highlighted.

## 1. Introduction

Hypercortisolism (HC), i.e., prolonged high serum concentration of cortisol, is associated with the development of Cushing’s syndrome (CS). It can be caused by increased endogenous production of cortisol or by extended treatment with glucocorticoids (GCs), used frequently in the management of inflammatory, autoimmune or allergic diseases [1]. CS is one of the most common endocrinopathies in adult dogs [2]. As in humans, there is a pituitary (ACTH-dependent) form, which is the most common, and an adrenal form, which occurs in about 15% of canine cases [2]. Unless neurological symptoms occur, the forms cannot be differentiated based on clinical signs, which can be severe and distressing for owners and dogs. The most common signs are polyuria and polydipsia, resulting in nocturia, excessive panting and exercise intolerance through massive muscle loss [2]. During the course of disease, dogs develop disorders similar to the human metabolic syndrome, with hyperlipidemia, hypertension and secondary diabetes mellitus being frequently observed [3,4,5]. Therefore, based on the similarity with human metabolic syndrome, dogs have been used as a model to study dyslipidemia, obesity and diabetes mellitus [3,6]. In fact, as a model for human GC metabolism, dogs offer greater advantages than other species, in particular rodents. Accordingly, in dogs, as in humans, cortisone is converted into the biologically active cortisol, while in rodents the active form is corticosterone [7]. Furthermore, based on the available literature, metabolic syndrome has not yet been described in rodents.

Adipose tissue is a complex organ, functioning not only as an energy storage site but also as an active endocrine organ. Thus, it secretes active peptides (like adipokines and cytokines) that are involved in the modulation of a wide variety of physiological processes, i.a., glucose and lipid metabolism, immune function, vascularization/angiogenesis and hormonal regulation [8,9]. Different types of adipose tissue are present in the body, e.g., subcutaneous, visceral, white and brown, diverging not only in their localization, but also in their embryonic origin (lineage), composition and/or function [10,11,12]. In fact, whereas chronic HC (either endogenous or iatrogenic) is associated with increased visceral adiposity, increased lipolysis and reduction of subcutaneous peripheral adipose tissue is observed [13,14,15]. This suggests the presence of divergent effects of GCs on different adipose tissues associated with fat tissue redistribution (culminating in central obesity) observed during CS [14]. In fact, under endogenous CS-associated hypercortisolemia, subcutaneous adipose tissue of humans presents increased lipolytic activity [16]. In contrast, GC signaling in the visceral adipose tissue (VAT) of mice is associated with increased adipocyte proliferation and expansion (adipogenesis), and modulation of glucose and lipid metabolism [17,18,19]. Following this, we hypothesize that chronic exposure to high cortisol levels might affect VAT also in dogs, especially concerning metabolic activity. Indeed, it is surprising that, despite the clinical importance of CS in canine patients, the effects of chronic HC on the adipose tissue of dogs have not yet been evaluated. Therefore, it is also unknown whether the changes within the adipose tissue seen in humans would similarly occur in dogs.

Considering the wide regulatory functions of adipose tissue, going beyond energy balance, the understanding of GC-induced changes in adipose tissue during a pathological process with high metabolic imbalances, such as CS, is crucial from the clinical point of view. Consequently, the aim of the present study was to examine the transcriptome of VAT in dogs under physiological and supraphysiological GC levels and to discuss the possible translational value of our findings. For this, an experimentally induced chronic hypercortisolemia model was used, followed by deep RNA sequencing (RNA-Seq) by applying next-generation sequencing (NGS) technology.

## 2. Materials and Methods

### 2.1. Animal Experiments and Tissue Sampling

Five Beagle dogs (3 males and 2 females) aged between 23 and 88 months and with body weights ranging from 14 to 19.8 kg were used in the present work. The health status of the animals was assessed by physical examination, blood count and biochemical analysis of blood and urine. Throughout the study, animals were housed as small groups in standard kennels in the facilities of the Vetsuisse Faculty, University of Zurich, Zurich, Switzerland, received adult maintenance dog food (Josidog Adult Sensitive, Josera, Kleinheubach, Germany), and had access to water *ad libitum*. All animal experiments were carried out in accordance with the guidelines determined by the Animal Welfare Act of Switzerland and approved by the Cantonal Veterinary Office of Zurich (permission number TVB 276/2014).

Hypercortisolemia and consequent iatrogenic hyperadrenocorticism were induced by the permanent administration of tetracosactide, the synthetic analog of ACTH (Bachem AG, Bubendorf, Switzerland) using osmotic pumps (Alzet osmotic pump model 2ML4, Durect, Cupertino, CA, USA) for 25 weeks, following our previous methodological description [20]. The osmotic pumps were implanted subcutaneously in the dorsolateral region of the neck and replaced every 4 weeks. Tetracosactide dosage started at 1.3–1.95 μg/kg/day, increasing to a final dosage of 6–10 μg/kg/day. After 25 weeks of treatment, hypercortisolemia was confirmed with a positive ACTH stimulation and low-dose dexamethasone suppression tests, as described previously [20].

For the collection of visceral adipose tissue (VAT), animals underwent general anesthesia and tissue was collected by laparotomy prior to treatment (Pre-ACTH samples) and after the 25 weeks of tetracosactide administration (Post-ACTH samples). Immediately after collection, adipose tissue (20–30 mg per dog) was snap frozen in liquid nitrogen and stored at −80 °C until RNA isolation.

### 2.2. RNA Extraction

Total RNA was extracted from visceral adipose tissue using the RNeasy Lipid Tissue Mini Kit (Qiagen GmbH, Hilden, Germany) following the protocol provided by the manufacturer. RNA purity and concentration were assessed with a NanoDrop 2000C spectrophotometer (ThermoFischer Scientific AG, Reinach, Switzerland), whereas sample integrity was evaluated by electrophoresis using the Agilent 2200 TapeStation system (Agilent, Waldbronn, Germany). RNA integrity number (RIN) of analyzed samples ranged between 7.6 and 7.8 and total RNA amounts between 732 and 1722 ng.

### 2.3. RNA Sequencing (RNA-Seq)

Quantitative evaluation of gene expression in all 10 samples, 5 pre-ACTH and 5 post-ACTH treatment, was performed by sequencing RNA (RNA-Seq) using next-generation sequencing (NGS) methodologies, as previously described [21]. Briefly, the quality and quantity of isolated RNA were further evaluated by resorting to Qubit (1.0) Fluorometer (Life Technologies, Carlsbad, CA, USA) and high-resolution automated electrophoresis in a Bioanalyzer 2100 (Agilent). A 260/280 ratio ranging from 1.8 to 2.1 and a 28S/18S ratio between 1.5 and 2.0 were required to proceed with sample analysis. Library preparation for each sample was performed using 100–1000 ng of total RNA and a TrueSeq RNA Sample Prep Kit v2 (Illumina, Inc., San Diego, CA, USA), and all samples were processed simultaneously to prevent batch effects. Due to low RNA amounts, samples were enriched by ribosomal RNA depletion. Afterwards, samples were reverse transcribed into double-stranded cDNA, fragmented and end repaired (polyadenylated) before being ligated with TruSeq adapters containing the multiplexing index. Then, selective enrichment of fragments containing TruSeq adapters on both ends was performed with PCR and the quality and quantity of libraries were evaluated with the Qubit (1.0) fluorometer and the Bioanalyzer 2100. Obtained libraries were finally normalized to 10 nM with Tris-Cl 10 mM containing 0.1% Tween 20.

Clusters were generated using 8 pM of pooled normalized libraries on the cBOT System using the TruSeq PE Cluster Kit v4-cBot-HS or the TruSeq SR Cluster Kit v4-cBot-HS, and sequencing was performed in an Illumina NovaSeq 6000 with single end 150 bps using the TruSeq SBS Kit (all obtained from Illumina, Inc., San Diego, CA, USA). The obtained raw data (fastq files) were used for the downstream analysis and were also deposited in the NCBI’s Gene Expression Omnibus, being accessible through the GEO Series accession number GSE178108.

### 2.4. Data Analysis

For initial analysis and quantitative assessment of gene expression, raw data were uploaded to the SUSHI framework [22,23], developed at the Functional Genomics Center of Zurich (FGCZ ETH/UZH, Zurich, Switzerland). Raw data were trimmed with Fastscreenapp and data quality was assessed using the CountQC. The obtained transcriptome dataset was aligned to the reference canine Ensembl genome build CanFam3.1 (http://www.ensembl.org/Canis_familiaris/Info/Index (accessed on 1 October 2020) using the Spliced Transcripts Alignment to a Reference (STAR) mapper. Gene expression was then quantified with the *featureCounts* function from the R package Rsubread [24], and a minimum of 10 reads were required to consider a gene as being expressed in a sample. For pairwise comparison, the contrast “post-ACTH over pre-ACTH”, using animal ID in addition to treatment as a pairing factor, was defined and the list of differentially expressed genes (DEGs) was obtained with the linear model DeSeq2 from the Bioconductor package [25], as previously described [21]. For the differential expression analysis, total number of mapped reads are normalized to size factors, representing sampling depth, estimated by DeSeq2 for each sample, after the *featurecounts*-computed normalization of the total number of mapped reads to the transcript length [24,25]. Afterwards, the Wald test was used to assess the significance of differential expression, and the Benjamini–Hochberg algorithm, which calculates the false discovery rate (FDR, adjusted *p*-value), was used for correction of multiple testing. The low yield of total RNA obtained from VAT precluded the validation of transcriptomic analysis through further mRNA quantification methodologies (e.g., semi-quantitative PCR). Thus, stringent thresholds of *p*-value and FDR, 0.01, were applied in the determination of DEGs. The complete list of DEGs used in downstream analysis is provided in Appendix A. Explorative analysis of the genes differentially expressed before and after ACTH treatment was performed by functionally characterizing up- and downregulated genes. The association of DEGs with gene ontology (GO) categories of biological processes was obtained with the Functional Annotation Tool from the Database for Annotation, Visualization, and Integrated Discovery (DAVID 6.8; https://david.ncifcrf.gov/ (accessed on 1 October 2020)) and further corroborated by analysis with the Bioconductor package *goseq* [26] and the online tool Pantherdb (http://pantherdb.org (accessed on 1 October 2020)) [27]. The identification and visualization of differentially enriched functional biological networks was performed with the ClueGO V2.5.1 application [28] for the open source bioinformatic platform Cytoscape V3.8.2 [29]. The prediction of the most significantly affected canonical pathways and identification of upstream regulators for the DEGs obtained was performed with Ingenuity Pathway Analysis software (IPA, Qiagen, Redwood City, CA, USA). Finally, the identification of significantly similar or different, and biologically relevant, transcriptomic analysis was performed with the function Analysis Match from IPA. This function matched shared patterns in, i.a., canonical pathways and upstream regulators, between the present dataset and other publicly available IPA analysis datasets using the Qiagen OmicSoft Suite. Lists with up to 20 representative genes and/or statistical details for different identified functional terms (GO, functional networks, canonical pathways, upstream regulators) are presented in Appendix A.

## 3. Results

### 3.1. Quality Control and Pairwise Comparison

Quality control of RNA-seq, as well as sample homogeneity and clustering were performed with the CountQC app provided in the SUSHI framework. Samples showed a high homogeneity within each group, and a treatment-dependent clustering of samples was obtained (Appendix A). This was further supported by the principal component analysis (PCA) plot for all genes identified (Figure 1). In this analysis, the distribution of samples in the principal component (PC) 1 was segregated by treatment. Furthermore, in the PC2, samples from the Post-ACTH group appeared to present a higher homogeneity than those from the Pre-ACTH group.

The effects of ACTH treatment in the transcriptome of VAT were evaluated through a differential expression analysis (pairwise comparison). For this, the contrast “Post-ACTH over Pre-ACTH” was defined in the DeSeq2 package for Bioconductor (provided in the SUSHI framework). From the 19,856 genes identified, 13,332 genes presented at least 10 reads in at least one sample, i.e., were considered as being expressed. For determining differentially expressed genes (DEGs) the threshold FDR < 0.01 was applied. Consequently, a total of 1190 DEGs were identified for the contrast “Post-ACTH over Pre-ACTH”, with 499 DEGs being upregulated and 691 downregulated (Figure 2). The full list of DEGs obtained, including statistical details, is presented in Appendix A. Further representation of the distribution of DEGs, filtered by FDR < 0.01, is presented as a volcano plot in Appendix A.

### 3.2. Functional Annotations

Functional classification of the DEGs was performed by determination of Gene Ontology (GO) terms related to Biological Processes using DAVID, enriched functional networks were identified with the ClueGO application for Cytoscape, and the predicted effects of treatment on the most significant canonical pathways for the present dataset, as well as prediction of upstream regulators, was accomplished with IPA. Lists of up to 20 representative genes for different functional terms and statistical details are provided in Appendix A.

#### 3.2.1. Gene Ontologies

Genes upregulated after 25 weeks of treatment with the ACTH analog were associated with translation (*p* = 4.33 × 10^−31^), ribosomal small subunit assembly (*p* = 3.50 × 10^−9^) and cytoplasmic translation (*p* = 9.43 × 10^−8^) (Figure 2, Appendix A). In contrast, the functional terms associated with genes downregulated after the ACTH-induced hypercortisolemia were (Figure 2, Appendix A): cell adhesion (*p* = 1.29 × 10^−6^), chemotaxis (*p* = 2.17 × 10^−6^), immune response (*p* = 8.05 × 10^−5^), collagen fibril organization (*p* = 1.37 × 10^−4^), positive regulation of cell migration (*p* = 1.45 × 10^−4^), integrin-mediated signaling pathway (*p* = 2.32 × 10^−4^) and angiogenesis (*p* = 1.75 × 10^−3^). These results were further corroborated by resorting to the package *goseq* from Bioconductor, and to the overrepresentation test provided in the Pantherdb online tool by using the canine genome as reference (Appendix A).

#### 3.2.2. Functional Networks

The visualization of biological terms in a functional grouped network was accomplished through the ClueGO plugin for the Cytoscape platform, using as input the lists of DEGs downregulated or upregulated in response to treatment. Networks overrepresented in the Post-ACTH group were mostly related to translation and ribosomal function (Figure 3A, Appendix A). In contrast, the most represented functional networks in genes downregulated after ACTH treatment were related to cell motility and migration, vascularization, immune signaling, cell–cell communication, intracellular signaling and extracellular matrix organization (Figure 3B, Appendix A).

#### 3.2.3. Canonical Pathways and Upstream Regulators

To predict the canonical pathways most significantly affected by treatment, as well as the possible upstream regulators involved in this response, the list of DEGs (*p* and FDR < 0.01) was uploaded to the IPA software. Among the pathways predicted to be activated after treatment were those related to translation (EIF2 signaling, *p* = 1 × 10^−26^; RhoGDI signaling, *p* = 5.01 × 10^−4^), as well as oxidative phosphorylation (*p* = 5.01 × 10^−4^), apelin adipocyte signaling pathway (*p* = 1.35 × 10^−2^) and IGF-1 signaling (*p* = 2.24 × 10^−2^) (Figure 4, Appendix A). In contrast, among the most enriched canonical pathways predicted to be deactivated after treatment were those related to (Figure 4, Appendix A): cellular proliferation and intracellular signaling (mTOR signaling, *p* = 1 × 10^−13^; regulation of eIF4 and p70S6K signaling, *p* = 3.16 × 10^−14^; signaling of Rho family GTPases, *p* = 5.01 × 10^−5^; protein kinase A signaling, *p* = 9.77 × 10^−4^; ephrin receptor signaling, *p* = 3.02 × 10^−3^; HGF signaling, *p* = 3.72 × 10^−3^), vascularization (adrenomedullin signaling pathway, *p* = 1.51 × 10^−5^; thrombin signaling, *p* = 3.73 × 10^−5^; endothelin-1 signaling, *p* = 2.88 × 10^−3^), immune function (leukocyte extravasation signaling, *p* = 2.51 × 10^−4^; IL8 signaling, *p* = 4.97 × 10^−4^; Tec kinase signaling, *p* = 4.37 × 10^−4^; Th2 pathway, *p* = 1.05 × 10^−2^; CXCR4 signaling, *p* = 1.1 × 10^−2^; chemokine signaling, *p* = 1.15 × 10^−2^), hormonal signaling (estrogen receptor signaling, *p* = 3.16 × 10^−4^, GNRH signaling, *p* = 2.82 × 10^−3^; relaxin signaling, *p* = 4.17 × 10^−3^), metabolism (phospholipase C signaling, *p* = 2.34 × 10^−6^; LXR/RXR activation, *p* = 4.79 × 10^−4^; superpathway of inositol phosphate compounds, *p* = 9.55 × 10^−4^; sphingosine-1-phosphate signaling, *p* = 3.31 × 10^−4^; insulin secretion signaling pathway, *p* = 4.79 × 10^−3^), HIF1α signaling (*p* = 1.38 × 10^−3^) and actin cytoskeleton signaling (*p* = 4.57 × 10^−3^). Furthermore, no prediction (undetermined z-score) on the effects of treatment over atherosclerosis signaling pathway (*p* = 2.29 × 10^−3^) could be achieved. 

Among the top upstream regulators predicted by the IPA software to affect the expression of the DEGs were factors such as the La ribonucleoprotein 1 translational regulator (LARP1, *p* = 1.62 × 10^−43^), tumor necrosis factor (TNF, *p* = 2.51 × 10^−28^), C-C motif chemokine receptor 2 (CCR2, *p* = 6.91 × 10^−15^), estrogen receptor 1/alpha (ESR1, *p* = 4.12 × 10^−14^) and interferon gamma (IFNG, *p* = 9.69 × 10^−15^), all predicted to be inhibited by treatment (Appendix A). In contrast, dexamethasone (*p* = 2.28 × 10^−32^) was predicted to be activated after ACTH treatment (Appendix A). Finally, no prediction of activation could be achieved for transforming growth factor 1 beta (TGFB1, *p* = 3.64 × 10^−28^), β-estradiol (*p* = 5.97 × 10^−27^), interleukin 1 beta (IL1B, *p* = 1.34 × 10^−14^) and fibroblast growth factor 2 (FGF2, *p* = 2.75 × 10^−14^) (Appendix A).

### 3.3. Normalized Counts of Representative Genes

Functional analysis highlighted the presence of different biological mechanisms modulated by ACTH treatment. As further evaluation of mRNA amounts (e.g., with semi-quantitative PCR) was not possible in the present work (discussed below), further exploration of the expression patterns of representative DEGs for different functional groups was performed based on normalized gene counts (*p* and FDR < 0.01, Figure 5). In response to treatment, genes involved in intracellular signaling (peptidase inhibitor 3, *PI3*; phospholipase C delta 4, *PLCD4*), in ribosomal structure (such as the ribosomal protein L, *RPL*, -*3* and -*8*), and the insulin receptor substrate 2 (*IRS2*) and insulin-like growth factor binding protein 2 (*IGFBP2*) presented significantly higher normalized counts than in the control samples. In contrast, growth factors, such as insulin-like growth factor 1 (*IGF1*) and fibroblast growth factor 1 (*FGF1*), presented lower normalized counts. With regard to the immune system, whereas the expression of the interleukin 1 receptor type 1 (*IL1R1*) was increased, the cluster of differentiation 86 (*CD86*), toll-like receptor 7 (*TLR7*), CCR2 and C-X-C motif chemokine receptor 4 (CXCR4) presented lower counts in response to ACTH. The number of replicates from vascular endothelial growth factor C (*VEGFC*) and of matrix metalloproteinase (*MMP*) -*2* was lower in ACTH-treated dogs, contrasting with increased amounts of endothelin converting enzyme 1 (*ECE1*), thrombospondin 1 (*THBS1*), tissue inhibitor of metallopeptidase 4 (*TIMP4*) and *MMP19*. Regarding the metabolism of glucocorticoids, hydroxysteroid 11-beta dehydrogenase (*HSD11B*) -*2* and glucocorticoid receptor (*NR3C1*/*GR*) availability were reduced by treatment, contrasting with increased availability of *HSD11B1*. Finally, whereas the availability of *ESR1* was decreased in hypercortisolemia, prolactin receptor (*PRLR*) availability was increased.

### 3.4. Analysis Match

Following functional analysis, the Analysis Match tool from IPA was used to compare the present analysis with the transcriptomic effects induced by different experimental designs or pathologies. The list of datasets obtained was filtered by the investigated tissue (visceral adipose tissue) and biological relevance for the current work.

This analysis identified the dataset GSE40231, derived from a multiorgan expression profiling study in humans presenting with atherosclerosis [30], as being significantly different from the current dataset. In that microarray analysis, by comparing the transcriptome of VAT with that of the liver, tissue-specific genes potentially involved in coronary arterial disease were identified [30]. When comparing the canonical pathways significantly affected in the present study and in humans with atherosclerosis, most were predicted to have contrary activation patterns (Figure 6): the canonical pathways phospholipase C signaling, integrin signaling, IL8 signaling, CXCR4 signaling, estrogen receptor signaling, GNRH signaling, signaling by Rho Family GTPases, HIF1α signaling, thrombin signaling and relaxin signaling were predicted to be deactivated in the VAT of dogs with Cushing’s syndrome but activated in humans with atherosclerosis. Exceptions were EIF2 signaling being activated and LXR/RXR deactivated in both datasets, despite different activation *Z*-scores, and oxidative phosphorylation that was activated in the present dataset, but no activation prediction could be made for human samples (Figure 6). Similarly, several of the top upstream regulators for both analyses presented a diverging predicted activation: GLI family zinc finger 1 (GLI1), CCR2, ESR1 and TGFB1 were predicted to be inhibited in samples from the present study but activated in the human dataset, whereas the peroxisome proliferator-activated receptor gamma coactivator 1 alpha (PPARGC1A), LDL receptor related protein 1 (LRP1), dexamethasone, estrogen receptor, NR3C1/GR and interleukin 1 receptor antagonist (IL1RN) activity presented an inverse tendency (Appendix A). In contrast, TNF, interferon regulatory factor 2 (IRF2) and interferon gamma (IFNG) were predicted to be inhibited and insulin activated in both datasets, and the computing of LARP1 activity was not possible in VAT samples from humans with atherosclerosis.

## 4. Discussion

Strong effects of GCs on adipose tissue remodeling, expansion and metabolic function—inducing insulin resistance and affecting glycolysis, lipolysis and lipogenesis—have been described in vivo in humans and rodents and in in vitro models [16,17,18,19,31,32]. The lack of knowledge of such effects in the dog, despite the importance of Cushing’s syndrome in this species, was an important incentive for the present study. By implanting long-term ACTH-releasing pumps, we could induce a state of chronic endogenous HC in dogs, as demonstrated clinically and further highlighted by positive endocrine tests (ACTH-stimulation test, low-dose-dexamethasone suppression test) [20]. Clinically, the dogs showed polyuria/polydipsia, a loss in body weight, muscle loss and, selected dogs, also an increased central obesity as described [20]. We were able to evaluate, for the first time in the dog, the effects of chronic endogenous HC on the transcriptome of VAT. Due to the low amounts of total RNA obtained from VAT, which was a consequence of the low amount of the available adipose tissue, the possibilities for downstream validation of NGS results by other quantitative methods was limited. Therefore, strict thresholds in determining DEGs were applied, including low *p* and false discovery rate (FDR) values (<0.01). By using healthy dogs with high homogeneity regarding breed (Beagle), genetic background and body condition and using these animals as their own controls, individual variation in output results was decreased. Furthermore, the high homogeneity between samples on each group suggests a low influence of other confounding variables, such as sex or age, in the present analysis. Importantly, the presence of dexamethasone, a synthetic glucocorticoid, among the top upstream activated regulators supports the conclusion that several of the transcriptomic effects observed were induced by chronic HC. Treatment with ACTH instead of the administration of a synthetic GC provides results more comparable to naturally occurring CS. In contrast, dexamethasone, a synthetic GC frequently used in HC-related studies, is already in its active form and cannot be further activated by HSD11B1. Conversely, the administration of ACTH increases endogenous cortisol concentrations, which then can be both activated by HSD11B1 and inactivated by HSD11B2 [33].

In accordance with our hypothesis, long-term ACTH exposure and consequent HC clearly affected the transcriptome of adipose tissue, as shown by the distinct separation between samples collected before and after treatment in the principal component analysis, with 1190 genes further identified as being differentially expressed (*p* and FDR < 0.01). Most identified functional terms were overrepresented in DEGs downregulated in response to chronic HC. Those involved, i.a., extracellular matrix, cell adhesion and migration, immune response, vascularization and intracellular signaling, and were accompanied by the predicted deactivation of most canonical pathways identified by IPA as significantly affected in the present dataset. As for functional terms overrepresented in treated samples, they were mainly related to translation and ribosome biogenesis, associated with apparently higher transcript counts of ribosome-related genes in response to treatment. This was further associated with the predicted activation of the canonical pathway signaling through EIF2, involved in the initiation of translation by binding tRNA to the ribosome [34]. Furthermore, LARP1, a RNA-binding protein that, among other mechanisms, regulates translation of several ribosomal subunits [35], was identified as the top-ranking predicted upstream regulator. All of these suggest an increased translational activity in visceral adipocytes after chronic HC induced by prolonged ACTH treatment. This appears to contrast with previous transcriptomic analysis of human subcutaneous adipose tissue, where most differentially expressed ribosomal subunits were downregulated in patients presenting with Cushing’s disease [32]. It is noteworthy that visceral and subcutaneous adipocytes arise from different cell lineages [10] and also present functional differences in the dog, as implied by the diverging expression patterns of lipogenesis-related factors observed in response to weight loss [11]. Thus, the differences in transcriptional signaling between the present work and human subcutaneous tissue appear to be tissue-specific rather than dependent on species and are possibly related to the fat tissue redistribution, with VAT expansion, observed during CS [14].

Different growth factors involved in the regulation of tissue expansion were also affected by ACTH treatment. The expression of insulin-like growth factor 1 (*IGF1*) was downregulated in hypercortisolemia. In fact, GCs appear to have a suppressive effect on IGF1 expression in adipose tissue, as shown by the significantly decreased expression of IGF1 in the periadrenal adipose tissue of human patients with CS [19]. Furthermore, in consonance with our observation of lower transcriptional availability of *FGF1* after treatment, FGF signaling was previously predicted to be deactivated in the periadrenal adipose tissue of humans with CS [19]. In gonadal white adipose tissue, FGF1 availability can be induced by a high-fat diet, and its knockout, in conjunction with the same high-fat diet, affects adipose expansion and dysregulates metabolic activity in mice [36,37]. This was further highlighted by the decreased circulating glucose levels observed after the administration of rFGF1 to insulin-resistant mice [38]. Furthermore, the decreased expression of FGF1, which acts as a proangiogenic factor [39], might also represent one of the vasomodulatory effects of treatment. Indeed, ACTH treatment was associated with the downregulation of *VEGFC* and upregulation of endothelin converting enzyme 1 (*ECE1*) and *THBS1*, with functional terms related to vascularization being enriched for genes downregulated after treatment. The ECE1 activates endothelins that modulate vascular dilation or constriction through its receptors ETA or -B [40]. Whereas the increased availability of this factor does not allow a prediction of endothelin activity changes per se (i.e., vasodilation or constriction), the effects observed in the number of transcripts of the proangiogenic VEGFC [41] and of the antiangiogenic *THBS1* [42] suggest that HC might disrupt angiogenic activity in visceral adipose tissue. In fact, VEGF activity was reported to be apparently affected in human periadrenal adipose tissue in patients with CS [19]. Furthermore, the decreased adiposity and insulin levels observed in THBS1-null mice, associated with immunomodulatory effects in visceral adipose tissue [43], suggest a broader role of THBS1 in adipose physiology that might be worth exploring in the dog.

Effects of treatment in the regulation of tissue structure could also be observed in the modulation of extracellular matrix organization, with the tissue inhibitor of metalloproteinases-4 (*TIMP4*), an inhibitor of, i.a., MMP2 (downregulated herein), presenting higher counts of transcripts after long-term HC. To the authors’ knowledge, this is the first time that the expression of these factors has been described in canine VAT. Among other functions, TIMP4 modulates the differentiation of adipocytes, being upregulated during the terminal phase of maturation of preadipocytes [44]. Furthermore, TIMP4-deficient mice present a lower accumulation of visceral fat, in a mechanism that involves CD36 signaling through PI3 [45], the latter being upregulated by treatment in the present study. TIMP4 was also downregulated in visceral adipose tissue of GR knockout mice [19], further highlighting the GC-dependency of the expression of this metalloprotease inhibitor. Thus, considering the present results and the mechanisms described in mice, it is possible that TIMP4, upregulated in response to HC and possibly regulating MMP2 availability, might be involved in the modulation of adipocyte growth and increased visceral fat observed in dogs presenting with CS.

At the metabolic level, the observed increased availability of *IRS2, IGFBP2* and, to a lower extent, *IRS1* after treatment suggested an elevated sensitivity to insulin in VAT under hypercortisolemia. Indeed, while CS is frequently associated with systemic insulin resistance, several studies in humans have indicated that such is not the case in adipose tissue, where GCs are associated with increased adipocyte expression of factors such as IR, IRS1 and IRS2, and glucose uptake both in vitro and in vivo [46,47,48]. This GC-induced increased insulin sensitivity appears to represent a compensatory mechanism, as suggested by others [49], where insulin signaling favors adipose expansion and lipid accumulation to compensate for increased lipidemia. Although interesting, unfortunately, insulin resistance was not evaluated in the present study. Nevertheless, the plasma lipase activity was significantly increased 25 weeks after ACTH treatment [20]. Moreover, plasma triglycerides concentrations (TG) were also significantly increased in these dogs [20]. Clinically, all dogs showed a significant loss in body weight, muscle loss, and in some dogs, increased central obesity was observed [20].

While the transcript counts of the lipid hydroxylase *PLCD4* increased with treatment in the present study, the predicted deactivation of lipid-related canonical pathways such as LXR/RXR signaling and sphingosine-1-phosphate signaling further supports the presence of lipid accumulation mechanisms in the canine VAT in response to hypercortisolemia, as also described in humans [50].

The induction of prolonged hypercortisolemia appeared to modulate GC signaling within canine VAT, with the decreased availability of *NCR3C1* (glucocorticoid receptor, GR) possibly representing a local compensatory mechanism to chronic GC excess. Surprisingly, that appeared not to be the case for both hydroxysteroid 11-beta dehydrogenases: *HSD11B1*, which catalyzes the conversion of cortisone into the biologically active cortisol [33], was upregulated while *HSD11B2*, which reverts cortisol activation [33], presented lower counts of transcripts after treatment. Despite requiring further validation, these expression patterns suggest an active increased availability of cortisol in canine VAT, even under hypercortisolemia conditions. With regard to their effects in adipose tissue, HSD11B1 global knockout in mice is associated with decreased adipose accumulation; more interestingly, specific knockout in adipose tissue resulted in a lower degree of hepatic steatosis and lower levels of circulating fatty acids after corticosterone administration [51]. Furthermore, HSD11B2 adipose-specific overexpression had a positive impact on the metabolic status of obesity-induced mice [52]. Thus, the dysregulation of local GC metabolism in VAT suggested by the present results, associated with an increased local availability of cortisol, might represent a treatment target in dogs with CS, being possibly involved in the regulation of adipocyte metabolism and expansion, and overall VAT accumulation in CS. Nevertheless, as this study was performed in vivo, some of the observed effects might not be directly induced by cortisol but, instead, could potentially constitute a response to other local or systemic signals, which have not been evaluated in the present study and therefore could be regarded a potential limitation of the study. Moreover, number of animals was rather low, meaning that factors like sex or age could not be assessed.

The modulation of hormonal receptors, i.e., ESR1 and PRLR, in response to treatment further highlighted the multifactorial effects of prolonged hypercortisolemia in visceral adipose tissue function. PRL signaling is associated with the regulation of adipogenesis [53], mainly in rodents, while an interplay between the GC receptor and ESR1 was previously shown in vitro regarding immunomodulatory activity [54]. Nevertheless, the role of such receptors in VAT, according to the authors’ knowledge, remains unexplored in the canine species.

Finally, induction of hypercortisolemia with ACTH was also associated with the modulation of immune activity in visceral adipose tissue. This was highlighted by the presence of enriched immune-related functional terms in samples collected after treatment and predicted deactivation of canonical pathways. The presence of immunomodulatory responses to hypercortisolemia is in agreement with the immunoregulatory, usually anti-inflammatory, effects of GCs in several systems and has been previously reported in humans presenting with Cushing’s syndrome [55,56,57,58]. Hypercortisolemia was previously associated with an increased infiltration of macrophages into VAT in humans [58]. In the present work, the decreased availability of *CD86*, expressed by some phenotypes of macrophages as M1 and M2b [59], suggests a local modulation of the immune milieu in response to HC. However, the evaluation of such changes is still required. Interestingly, decreased counts of *CCR2* and *CXCR4* also occurred after prolonged ACTH treatment. Among other functions, these two chemokine receptors have been implicated in vascular damage and formation of atherosclerosis [60,61]. In humans, hypercortisolemia is associated with an increased risk of developing atherosclerotic lesions [62]. In a retrospective study, the presence of metabolic diseases such as diabetes mellitus and hypothyroidism in dogs was associated with a higher risk of development of atherosclerosis, but not Cushing’s syndrome [4]. This was a surprising observation, as both diseases, diabetes mellitus and hypothyroidism, are associated with hyperlipidemia, comparable to that seen in dogs with CS. Furthermore, using the analysis match tool from IPA, the transcriptome of peritoneal adipose tissue of humans presenting with atherosclerosis contrasted significantly with the results of the present work. The IPA analysis tool compares the investigated dataset with over 80,000 publicly available IPA analyses. The fact that only one existing analysis could be identified underlines the lack of knowledge on the involvement and function of VAT in metabolic alterations under HC and need for further investigations. Furthermore, as no analysis of the VAT transcriptome patients presenting with CS was available, the aim of this comparison was to assess possible functional and/or metabolic local mechanisms that could be associated with the development of atherosclerosis. Interestingly, most canonical pathways showed an inverse predicted activation pattern compared with the dog. Whereas this phenomenon still remains to be better explored in dogs, species-specific functional differences in the response to hypercortisolemia, probably involving, i.a., CCR2 and CXCR4, may be involved in the absence of increased atherosclerosis in these animals, contrasting with other metabolic diseases in dogs and with what is observed in humans. 

## 5. Conclusions

The present work provides the first insights into the transcriptome of canine visceral adipose tissue (VAT) in response to ACTH-induced chronic HC. This analysis allowed the identification of factors and potential mechanisms that might play important roles in the modulation of VAT in response to HC and possibly included targets for clinical applications. The modulation of local metabolism of GCs, with the HSD11B1/-2 expression patterns suggesting an increased local availability of cortisol, may comprise an important target in the modulation of GC-induced effects in VAT. Beyond our hypothesis, effects of treatment were clearly multifactorial and, in part, appeared to be associated with an increased expansion of VAT involving the modulation of growth factors such as *IGF1* and *FGF1,* ECM organization through *TIMP4*, and lipidosis. All these effects were similar to what has been described in humans. Thus, similarities of CS in the dog and humans, besides involving the previously described metabolic and clinical features, appear to include effects exerted upon visceral adipose tissue. These similarities further suggest the dog as a possible model for human CS. Moreover, vascularization, immune activity (possibly involving immunosuppressive signals) and sensitivity of VAT to different hormones were also affected by treatment. However, the full nature of these effects involving vascular and immune responses, as well as their relation to the situation observed in humans presenting with CS, could not be established in the present study. Thus, species-specific mechanisms, including, e.g., the observed discrepancies in the atherosclerosis development pathway involving CCR2 and CXCR4 signaling, should be taken into account in future research. 

## Figures and Tables

**Figure 1 vetsci-09-00250-f001:**
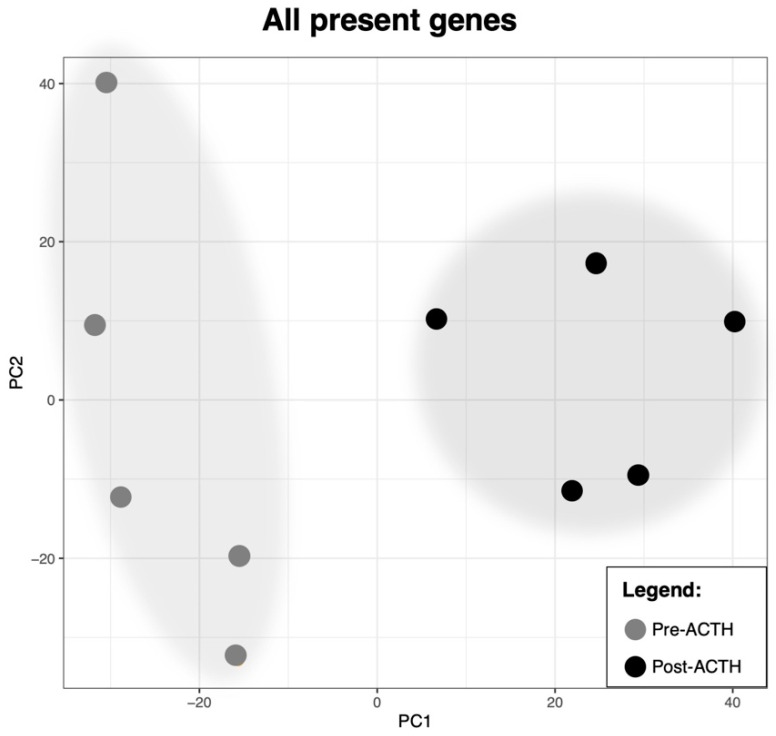
Principal component analysis (PCA) plot of all samples related to all detected transcribed genes. Sample distribution following the principal component 1 (PC1) appears to be prompted by the presence or absence of treatment.

**Figure 2 vetsci-09-00250-f002:**
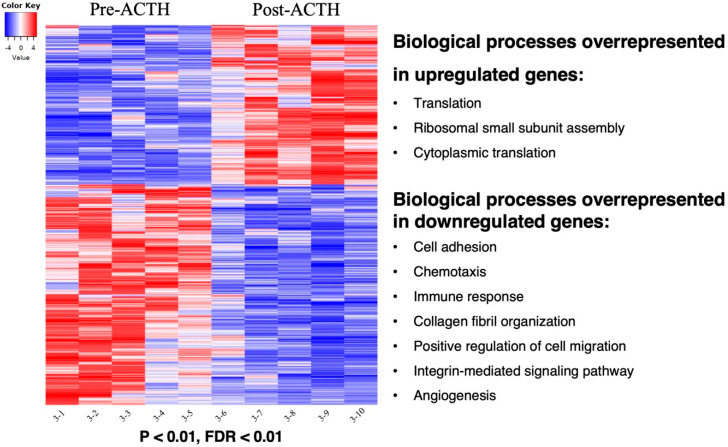
Heatmap of the 1190 differentially expressed genes (DEGs) in response to ACTH treatment and related functional terms analysis. Gradient of expression of each gene relative to its average expression is represented by colors red (high) to blue (low). 499 genes were upregulated in visceral adipose tissue (VAT) of animals after ACTH treatment whereas 691 genes were downregulated in samples before ACTH treatment (*p* < 0.01, FDR < 0.01). Main significant overrepresented gene ontologies (biological processes) identified with DAVID analysis in each group of genes are presented. The entire list of DEGs and details of functional terms can be found in Appendix A, respectively.

**Figure 3 vetsci-09-00250-f003:**
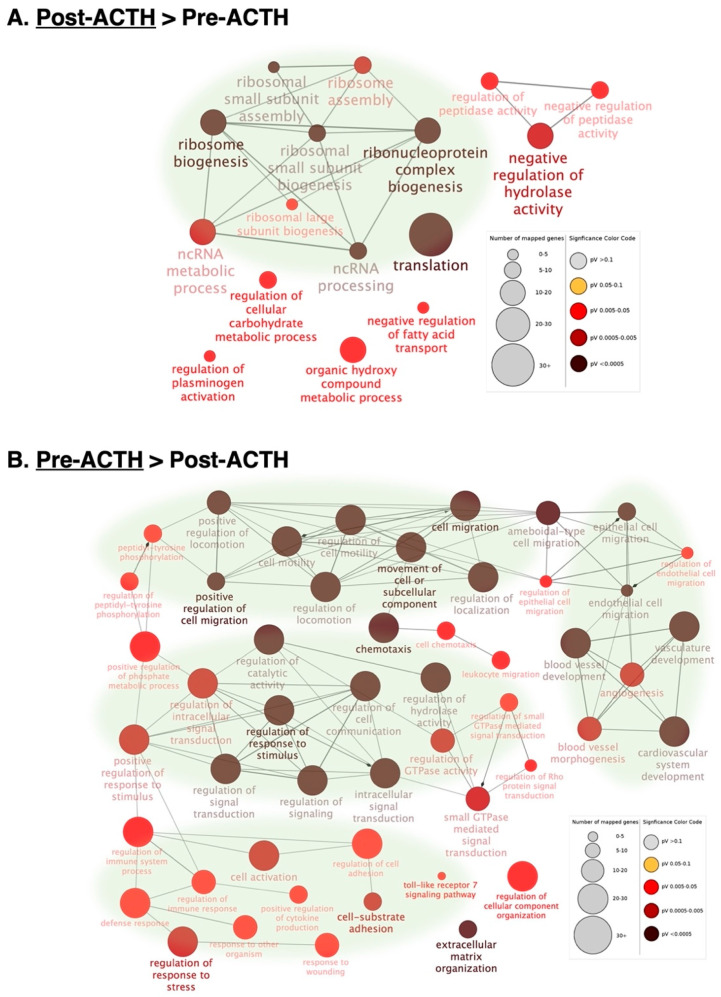
Functional networks as determined by ClueGO (Cytoscape) found in differentially expressed genes (DEGs) affected by ACTH treatment. Overrepresented functional terms are shown for genes upregulated (**A**) and downregulated (**B**) after ACTH treatment. Redundant and/or noninformative terms were removed and obtained networks were rearranged manually. Node size indicates number of mapped genes and level of enhancement is denoted by node color (represented in figure legend). Networks more highly represented after treatment were related to transcription whereas networks related to cell motility, vascularization, cell–cell communication and intracellular signaling, immune signaling and extracellular matrix were more highly represented in samples from the pre-treatment group. The list of representative genes included in different nodes and statistical details are listed in Appendix A.

**Figure 4 vetsci-09-00250-f004:**
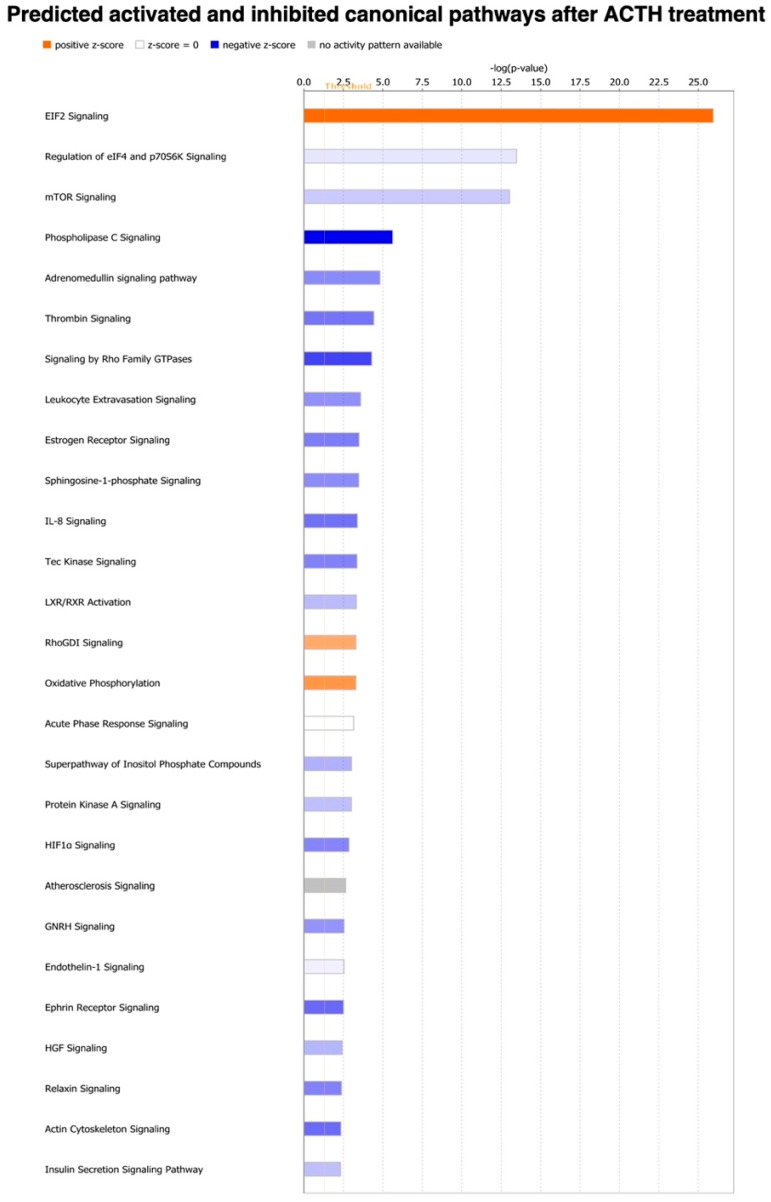
Top canonical pathways enriched in adipose tissue following treatment with ACTH. Pathways were ranked according to −log (*p* value). The predicted activity pattern of each given pathway after ACTH treatment is indicated by color (as defined in figure legend) and follows the calculated *Z*-score: positive *Z*-score indicates activation while negative *Z*-score relates to predicted deactivation of canonical pathway. Canonical pathways considered not related to the analyzed tissue or treatment were removed. All DEGs (*p* and FDR < 0.01) were used in the present analysis. List of selected canonical pathways, statistical details and representative genes included on each pathway are described in Appendix A.

**Figure 5 vetsci-09-00250-f005:**
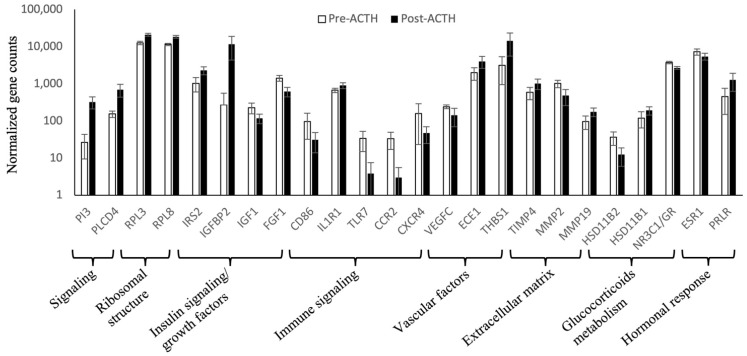
Graphical representation of normalized counts of selected genes. Factors involved in different biological mechanisms were selected and the data present mean ± SD of normalized counts of each gene in all samples from each group on a logarithmic scale. All factors were identified as being DEGs (*p* < 0.01, FDR < 0.01) by the DeSeq2 linear model.

**Figure 6 vetsci-09-00250-f006:**
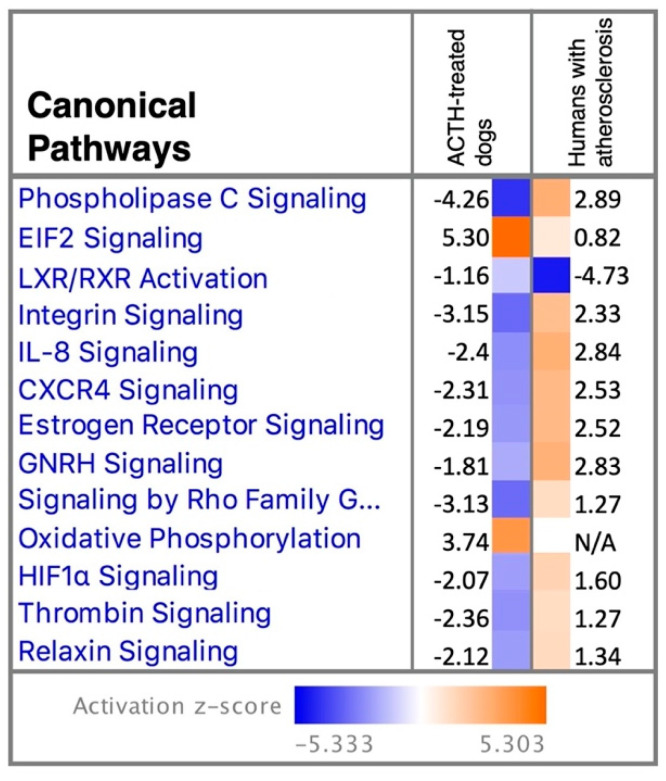
Main selected canonical pathways enriched in visceral adipose tissue (VAT) of ACTH-treated dogs detected in the present study and of humans presenting with atherosclerosis. The analysis match feature from the IPA software was used to compare enriched pathways in the transcriptome of VAT of dogs treated with ACTH (induced Cushing’s syndrome) with enriched pathways in the microarray of VAT of humans presenting with atherosclerosis [30]. Presented pathways were ranked according to *Z*-score. The predicted activity pattern of each given pathway is indicated by color (as defined in figure legend) and follows the calculated *Z*-score (positive *Z*-score indicates activation while negative *Z*-score relates to predicted deactivation of canonical pathway). Generally, an inverse predicted behavior of presented pathways was observed between the compared analyses.

## Data Availability

Raw data files (.fastq files) are publicly available in NCBI’s Gene Expression Omnibus with the GEO Series accession number GSE178108.

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
