# Peer review of "Effects of ACTH-Induced Long-Term Hypercortisolism on the Transcriptome of Canine Visceral Adipose Tissue"

_vetsci, 2022, doi:10.3390/vetsci9060250_

Round 1

Reviewer 1 Report

The main question addressed by the research is the transcriptomic effects of chronic hypercortisolemia in visceral adipose tissue of dogs were evaluated.  Compared with other published material it add to the subject area about he local modulation of visceral adipose during chronic hypercortisolemia presents interesting clinical goals of treatment.

I consider that the methodology used in this study is adequate.

In my opinion, the limitations of the study should be expanded.

I believe that proposals for the future should be included

The figures must have higher quality

Reviewer 2 Report

In this manuscript, Authors presented their studies on the effects of long-term hypercortisolemia (HC) on the transcriptome of canine visceral adipose tissue. Starting with the knowledge that Cushing's syndrome (CS) is a common endocrinopathy in adult dogs that is similar to CS in humans in its phenotypic and biochemical manifestations, Authors highlight the current lack of knowledge of the pathogenetic mechanisms leading to the dyslipidemia and obesity that frequently accompany CS in both humans and dogs.

Considering the importance of the broad regulatory function of adipose tissue, understanding the changes that occur during CS is clinically critical. In addition to in vivo studies in humans and rodents and in vitro models of the effects of glucocorticoids on adipose tissue remodeling, Authors propose their in vivo model of CS in dogs as an effective model to study the metabolic syndrome not only in canine CS but also in human CS. By implanting long-term ACTH-releasing pumps, Authors induced a state of chronic endogenous HC in dogs. In a previous study, they demonstrated that dogs exhibited the CS phenotype both biochemically and clinically. With the present study, Authors evaluated the VAT transcriptome in this model. Interestingly, they compared the VAT transcriptome before and after ACTH treatment in the same dogs to identify differentially expressed genes. As stated in the conclusions, the study leads to the identification of factors and potential mechanisms that may play important roles in the modulation of VAT in response to HC possibly indicating targets for clinical applications.

 The manuscript is well written and easy to understand. The introduction clearly stated the background and purpose of the study. The experimental design is well described, including material and methods. The data analysis included a number of bioinformatics and statistical tools that make the results reliable. The results are clearly presented and discussed.

Reviewer 3 Report

This manuscript reports a comprehensive study into the transcriptome of visceral adipose tissue in dogs in which chronic hypercortisolemia was induced by treatment with ACTH. The data presented are extensive and indicate that, effectively, a model of Cushing’s Syndrome was developed in the dogs.

I feel that the study has been well done and reported. I have a few comments that the authors may consider in any revision of their manuscript.

While the discussion of the findings is comprehensive, and the possibility of the dog as a model of Cushing’s Syndrome in humans is apparent, I would like to have seen the findings placed into more context physiologically. How are the findings relevant to the onset and physiological regulation of Cushing’s Syndrome beyond simply presenting potential targets for further clinical studies? I feel that the development and presentation of a hypothesis in the Introduction would assist in dealing with this point.

Was sex taken into consideration in the analyses? Sex differences in the functioning of the hypothalamo-pituitary adrenal axis have been well documented in a range of species.

The number of subjects in this study was small. Was this adequately accounted for in the analyses?

The terminology for glucocorticoid receptor needs to be clarified in the manuscript. Glucocorticoids bind to mineralocorticoid receptors, also termed glucocorticoid receptor I, and glucocorticoid receptors, also termed glucocorticoid receptor II.

Please note that the word data is plural, so it should be data “were” uploaded and date “were” trimmed.

Reviewer 4 Report

The authors present the manuscript “Effects of ACTH-induced long-term hypercortisolemia on the transcriptome of canine visceral adipose tissue” which describes RNA-seq data of VAT in dogs before and after ACTH treatment. As non-native speaker this manuscript seems well written. The authors however made some huge mistakes which might change the entire story.

My major concern is the fact that samples before and after treatment are paired samples, which is completely ignored in the analysis. Genetic background greatly influences transcriptome. These 5 dogs however have been considered as one group of controls and after treatment as an independent group of cases.

One other major thing is the thresholding for genes that might or might not be expressed. The human testis is reported to be the tissue containing the most expressed genes (~15.000). This study detects a stunning 19856, which is close to the full transcriptome. The number of DEGs is also fairly high comprising about 5% of the full transcriptome. This might be due to the loose thresholds that define expression of a gene. The risk of such a descriptive study is the lack of opportunities to confirm these findings. Several human VAT transcriptomes are available through GEOprofiles for comparative genomics.

These two major choices will reflect on the resulting data and since this is a descriptive study, it will therefor possibly change the entire storyline.

A third point of improvement is the mapping. Currently, mapping was performed using CanFam3.1. Currently, improved genomes are available, in particular CanFam6 or ROS1.0. This will reduce false positive and false negative calls.

The following other things caught my attention:

Title: hypercortisolemia should be called hypercortisolism according to ALIVE guidelines.

Bit of a messy abstract: the abbreviation GC is used once, but not written in full. RNA-Seq is introduced as abbreviation, but not used more than once. HC is introduced in the beginning of the abstract, but not consistently used throughout the entire abstract.

The keywords are identical to what is mentioned in the title, this will not help you to gain more visibility.

Figure 2 admittedly is a very colorful figure, however is not of additive value.

In paragraph 3.2.3 rounding up the p-values is performed inconsistently and reads very messy. These data should rather be presented in a table, which should replace figure 4.

Paragraph 3.3 mentions normalized counts, however, normalization is not described in the methods section.

Figure 5 lacks a y-axis legend

I honestly have no clue what the additive value of paragraph 3.4 would be. Different organisms, pathology, cell type etc give different gene signatures. If this is the message, then I am convinced, but I would discard this whole section.

Round 2

Reviewer 4 Report

I want to thank the authors for the adapted version of the manuscript and for answering the raised questions. By clarifying these, the manuscript is fit to be published in Veterinary Sciences.

Author Response

The authors thank again the reviewer for his support to improve the manuscript, which now seems acceptable for publication.